# Increased Creatine Kinase May Predict A Worse COVID-19 Outcome

**DOI:** 10.3390/jcm10081734

**Published:** 2021-04-16

**Authors:** Daniele Orsucci, Michele Trezzi, Roberto Anichini, Pierluigi Blanc, Leandro Barontini, Carlo Biagini, Alessandro Capitanini, Marco Comeglio, Paulo Corsini, Federico Gemignani, Roberto Giannecchini, Massimo Giusti, Mario Lombardi, Elena Marrucci, Alessandro Natali, Gabriele Nenci, Franco Vannucci, Gino Volpi

**Affiliations:** 1Unit of Neurology, San Jacopo Hospital, 51100 Pistoia, Italy; gino.volpi@uslcentro.toscana.it; 2Unit of Neurology, San Luca Hospital, 55100 Lucca, Italy; 3Unit of Infectious Diseases, San Jacopo Hospital, 51100 Pistoia, Italy; michele.trezzi@uslcentro.toscana.it (M.T.); pierluigi.blanc@uslcentro.toscana.it (P.B.); 4Unit of Diabetology, San Jacopo Hospital, 51100 Pistoia, Italy; roberto.anichini@uslcentro.toscana.it; 5Intensive Care Unit, San Jacopo Hospital, 51100 Pistoia, Italy; leandro.barontini@uslcentro.toscana.it; 6Unit of Geriatry, San Jacopo Hospital, 51100 Pistoia, Italy; carlo.biagini@uslcentro.toscana.it; 7Unit of Nephrology, San Jacopo Hospital, 51100 Pistoia, Italy; alessandro.capitanini@uslcentro.toscana.it; 8Unit of Cardiology, San Jacopo Hospital, 51100 Pistoia, Italy; marco.comeglio@uslcentro.toscana.it; 9Department of Laboratory Medicine, San Jacopo Hospital, 51100 Pistoia, Italy; paulo.corsini@uslcentro.toscana.it (P.C.); roberto.giannecchini@uslcentro.toscana.it (R.G.); 10Department of Internal Medicine, San Jacopo Hospital, 51100 Pistoia, Italy; federico.gemignani@libero.it (F.G.); massimo.giusti@uslcentro.toscana.it (M.G.); e_marrucci@yahoo.it (E.M.); gabriele.nenci@uslcentro.toscana.it (G.N.); 11Unit of Gastroenterology, San Jacopo Hospital, 51100 Pistoia, Italy; mario.lombardi@uslcentro.toscana.it (M.L.); alessandro.natali@uslcentro.toscana.it (A.N.); 12Unit of Pulmonology, San Jacopo Hospital, 51100 Pistoia, Italy; franco.vannucci@uslcentro.toscana.it

**Keywords:** CK, coronavirus, CPK, myopathy, SARS-CoV-2

## Abstract

Early reports from Asia suggested that increased serum levels of the muscular enzyme creatine-(phospho)-kinase (CK/CPK) could be associated with a more severe prognosis in COVID-19. The aim of this single-center retrospective cohort study of 331 consecutive COVID-19 patients who were hospitalized during Italy’s “first wave” was to verify this relationship, and to evaluate the role of possible confounding factors (age, body mass index, gender, and comorbidities). We subdivided our cohort in two groups, based on “severe” (*n* = 99) or “mild” (*n* = 232) outcomes. “Severe” disease is defined here as death and/or mechanical invasive ventilation, in contrast to “mild” patients, who were discharged alive with no need for invasive ventilation; this latter group could also include those patients who were treated with non-invasive ventilation. The CK levels at admission were higher in those subjects who later experienced more severe outcomes (median, 126; range, 10–1672 U/L, versus median, 82; range, 12–1499 U/L, *p* = 0.01), and hyperCKemia >200 U/L was associated with a worse prognosis. Regression analysis confirmed that increased CK acted as an independent predictor for a “severe” outcome. HyperCKemia was generally transient, returning to normal during hospitalization in the majority of both “severe” and “mild” patients. Although the direct infection of voluntary muscle is unproven, transient muscular dysfunction is common during the course of COVID-19. The influence of this novel coronavirus on voluntary muscle really needs to be clarified.

## 1. Introduction

Several neurological and neuromuscular symptoms have been identified as part of the COVID-19 spectrum, including muscle pain and fatigue. [1] Early reports from Asia, discussed elsewhere, [2] have suggested that increased serum levels of the muscular enzyme creatine (phospho)-kinase (CK or CPK) could be linked with a worse prognosis. This observation has been further confirmed by most recent retrospective studies [3] and systematic reviews [4,5]. Rare case reports from Europe are also available [6]. However, data on this marker of muscular damage have only briefly been mentioned in most papers. Even if some association between CK levels and the clinical outcomes of patients infected by SARS-CoV-2 seems to exist, the precise mechanisms are still unknown. Furthermore, it is not known if this observed relationship is true or if it is merely a spurious association caused by other clinical or demographic factors. The aim of this retrospective cohort study was to confirm (or exclude) this relationship in a Caucasian population, and to evaluate the role of possible confounding factors such as age, body mass index (BMI), gender and associated disorders.

## 2. Patients and Methods

This was a retrospective, single-center cohort study. We reviewed the clinical and laboratory data of 331 consecutive COVID-19 patients (156 females and 175 males) who were hospitalized during Italy’s “first wave” (March–May 2020) in the area of Pistoia, Tuscany (San Jacopo Hospital). These data were used to evaluate: (1) clinical outcomes (“severe” disease is defined here as death and/or mechanical invasive ventilation, in contrast to “mild” patients, who were discharged alive with no need for invasive ventilation; this latter group could also include those patients who were treated with non-invasive ventilation), (2) the CK levels evaluated at admission with standard methods, and expressed both as a continuous variable (U/L) and as a dichotomic variable (hyperCKemia: CK >200 U/L, which is the upper limit of normal in our laboratory, not different from that in previous studies on COVID-19 [7]), (3) basic clinical features potentially able to act as confounding and modifier factors (i.e., BMI, gender, and age), and (4) comorbidities.

Statistical analyses were conducted using R 4.0.3 x64. Most of the continuous variables (e.g., CK levels, patients’ age, and BMI) were not normally distributed and, therefore, were compared by the unpaired Wilcoxon rank sum test with continuity correction or correlated by Spearman’s test. Pearson’s Chi-squared test was used for categorical associations (if not differently specified). Given a two-sided alpha level of 0.05 and a power of 80%, we calculated that a sample of at least 72 subjects/group would be sufficient for the association analyses, in order to identify a difference of at least 0.2 (i.e., 30% versus 10%). Bonferroni’s correction for multiple tests was applied where appropriate. Statistical significance was set at a two-tailed *p* value of 0.05. Finally, multiple logistic regression was applied for testing a predictive equation combining the significant variables, and for evaluating if CK was an independent predictor.

For the discussion, PubMed was searched up to February 8th, 2021, for all articles in English about “(sars* OR covid* OR coronavirus) and (CK OR CPK OR creatine kinase OR creatine phosphokinase OR muscular OR muscle)”, and the abstracts were reviewed to identify all the relevant publications.

## 3. Results

### 3.1. CK Levels

We subdivided our cohort in two groups of “severe” (death and/or mechanical invasive ventilation, *n* = 99) and “mild” (discharged alive with no need for invasive ventilation, *n* = 232) COVID-19 patients. Where available, hyperCKemia (>200 U/L) was observed in 16/111 (14.4%) of the mild patients and in 26/74 (35.1%) of the severe patients, with a highly significant *p* = 0.001.

The absolute CK levels were also higher in the severe group (mean, 259; median, 126; range, 10–1672 U/L) compared to the mild group (mean, 141; median, 82; range, 12–1499 U/L); *p* = 0.01 (Figure 1).

Considering the basic features commonly linked to more severe courses of the disease, the CK levels were influenced by gender, but not by age or BMI (see the next paragraphs).

Where CK follow-up data were available, they returned to normal values during hospitalization in 20/28 (71.4%) of the patients who had hyperCKemia at admission. This applies to both subjects with severe (14/21, 66,7%) and mild (6/7, 85.7%) COVID-19 courses (non-significant difference (n.s.), Fisher’s test).

### 3.2. Age

As expected, the clinical outcome was markedly worse in older subjects; i.e., in our severe COVID-19 group, 46/99 (46.5%) were older than 80 years, whereas only 56/232 (24.1%) of the mild patients were 80 years old or younger (*p* = 0.00006).

The CK levels were not significantly correlated with age (Spearman’s rho, 0.134; *p* = 0.07), and hyperCKemia was not significantly more frequent in those subjects older than 80 years (12/45, 26.7%) than younger ones (30/140, 21.4%).

### 3.3. Gender

Regarding gender, males had an increased proportion of “severe” outcomes than females (64/175, 36.6%, versus 35/156, 22.4%, *p* = 0.005). CK levels were more frequently increased (>200 U/L) in men (31/107, 29.0%) than in women (11/78, 14.1%); *p* = 0.02. Regarding absolute levels, the difference between males (median, 109; range, 10–1672 U/L) and females (median, 86.5; range, 10–1499 U/L) did not reach statistical significance.

### 3.4. Body Mass Index

BMI was not correlated with CK levels (Spearman’s rho, 0.0727; *p* = 0.36).

Furthermore, in our cohort, the BMIs were not significantly different between severe and mild cases (the median BMI was 26 in both groups).

The fact that increased BMI was not linked to a more severe prognosis may be, at least partly, explained by the negative correlation between BMI and age that we observed in our cohort (rho, −0.239; *p* = 0.00003).

### 3.5. Comorbidities

Considering a critical *p-*value of 0.0056 after Bonferroni’s correction for multiple (= 9) analyses, several comorbidities (heart failure, atrial fibrillation, diabetes, active cancer, tabagism, dementia and other chronic neurological conditions) were not associated with the outcome of COVID-19; however, chronic kidney disease (*p* = 0.000004), arterial hypertension (*p* = 0.00004) and ischemic heart disease (*p* = 0.0002) were strongly associated with a worse outcome. Age was markedly higher in patients with all of these three conditions (*p* ≤ 0.0001). However, the CK levels were not significantly different in patients with or without hypertension, chronic kidney disease or ischemic heart disease (n.s.). Therefore, these comorbidities do not appear to act as confounding factors for our analysis.

### 3.6. Combined Analysis

Multiple logistic regression (“Y” = outcome, severe versus mild) was applied for testing a predictive equation combining the “X” variables that, as described in the previous paragraphs, were found to be linked with the outcome (age, sex, chronic kidney disease, arterial hypertension, ischemic heart disease, and CK levels). The constructed model had a highly significant *p* = 0.0000002 (analysis of deviance), and CK levels turned out to be an independent predictor with a *p* = 0.05 (Table 1).

Of note, whereas hypertension and ischemic heart disease were not independent predictor variables in this model, the association between chronic kidney disease and a severe outcome was very strong and not merely explained by increasing age: in our cohort, the odds ratio was 8.59 (95% confidence interval, 1.75–42.10, *p* = 0.008).

When the same analysis was performed considering CK as a dichotomic variable (hyperCKemia: CK >200 U/L), the model still had a very high statistical significance (*p* = 0.00000007) and hyperCKemia was an independent predictor of severe outcomes (odds ratio, 2.70; 95% confidence interval, 1.24–5.86, *p* = 0.01).

## 4. Discussion

The results of this study, together with early reports from Asia (especially from China) [2] and more recent studies [3,4,5], confirm that CK levels at admission are higher in COVID-19 patients who later experience more severe outcomes, and hyperCKemia is associated with a worse prognosis in a Caucasian population as well.

We are aware that our study presents the limitations typical of all retrospective studies. A weakness of this work is the lack a more detailed clinical evaluation. For instance, anamnestic data about subtle symptoms such as fatigue and muscle pain were not systematically collected, precluding any analysis including these complaints. Furthermore, other pieces of information are not available (e.g., eventual intramuscular therapies or heavy physical efforts before hospitalization, laboratory data on CK isozymes and macro-CK, cytokine levels, etc.).

However, our study did not appear to be under-powered, and the influence of various confounding factors was evaluated. For instance, chronic kidney disease, arterial hypertension and ischemic heart disease, closely linked to an older age, were also associated with a “severe” outcome of COVID-19. The association between chronic kidney disease and a severe outcome was very strong and not merely explained by increasing age. CK levels were not significantly different in patients with or without these comorbidities, which, therefore, do not appear to act as confounding factors for our analysis. BMI was not correlated with CK and was not associated with a more severe prognosis in our cohort (likely because it was inversely correlated with age).

The clinical outcome was markedly worse in older subjects and in males. However, regression analysis confirmed that increased CK was a predictor for a “severe” outcome, independently of the other considered variables.

It is not clear to date if increased CK levels in COVID-19 patients are caused by true myopathic damage. Muscle pain and fatigue are common in both mild and severe cases [1]. An early study from Wuhan [7] reported that “skeletal muscle injury” (defined in the following way: “when a patient had skeletal muscle pain and elevated serum CK”, greater than 200 U/L) was significantly more frequent in severe COVID-19, compared to less severe diseases (19% versus 5%). The median CK levels were higher in the more severe group. Interestingly, patients with “muscle injury” had multiorgan damage, including more severe liver and kidney abnormalities [7].

Furthermore, rhabdomyolysis is a known cooccurrence [8] that might represent a contributing factor for adverse outcomes in some COVID-19 patients [9].

It has been postulated that myocytes are susceptible to direct muscle invasion by SARS-CoV-2 because the viral receptor angiotensin converting enzyme 2 (ACE2) is expressed in skeletal muscle. [10] In a woman deceased due to COVID-19, the histological examination of skeletal muscle revealed fibrin microthrombi, perimysial microhemorrhages, and adjacent muscle fiber vacuolar degeneration and necrosis. Electron microscopy revealed “cytoplasmic clusters of virus-like structures in degenerated cells”, but surprisingly, SARS-CoV-2 genomes were not researched in this tissue. Furthermore, even more surprisingly, it was not specified if these degenerated cells were myocytes [11].

It remains unclear whether hyperCKemia is due to a virus-triggered inflammatory response or direct muscle toxicity, and it must be noted that hyperCKemia may occur at similar frequencies in COVID-19 and influenza infection [12]. In our patients, increased CK was generally a transient phenomenon, which returned to normal values during hospitalization in the majority of both “severe” and “mild” patients.

These latter observations do not support a specific role for SARS-CoV-2 in causing direct viral myositis in most cases. However, SARS-CoV-2 may directly infect cardiomyocytes in some instances [13], and the possibility of direct viral myositis in rare cases cannot be ruled out. It is more likely that, in most patients, virus-induced systemic inflammation may evoke a catabolic response in voluntary muscle, resulting in muscular complaints, transient hyperCKemia and, later, muscle atrophy. Hypothetically, “myotoxic” cytokines (such as CXCL-10, IFN-γ, IL-1β, IL-6, IL-17, and TNFα) may be involved, [14] but specifically designed studies are needed to elucidate the precise mechanisms.

## 5. Conclusions

Although the direct infection of voluntary muscle is uncertain or rare (and still unproven), transient muscular dysfunction is common during the course of COVID-19. Furthermore, there is an independent relationship between the CK levels at admission and the clinical outcome. In our opinion, CK should be measured in all COVID-19 patients admitted to hospital. The influence of this novel coronavirus on voluntary muscle really needs to be clarified.

## Figures and Tables

**Figure 1 jcm-10-01734-f001:**
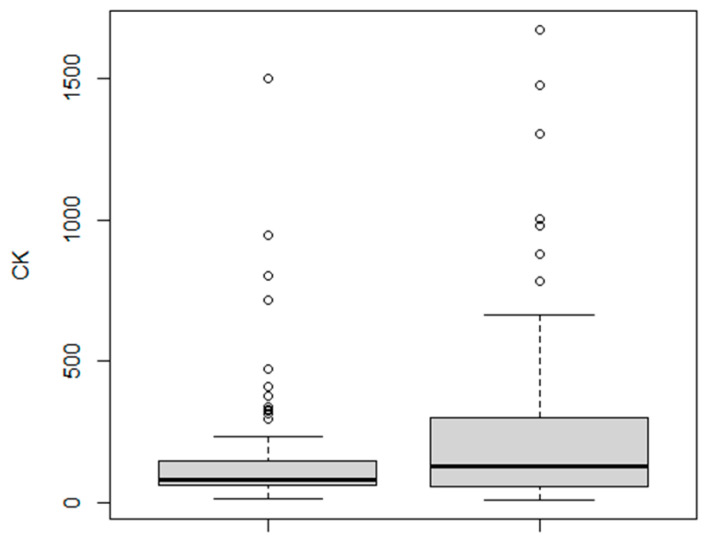
Higher CK levels in patients with “severe” (right) compared to “mild” (left) COVID-19. CK, creatine kinase, U/L.

**Table 1 jcm-10-01734-t001:** Multiple logistic regression.

	Estimate	Standard Error	z-Value	Odds Ratio	*p* (>|z|)
(Intercept)	−4.1139269	1.0706395	−3.842	0.0163	0.0001
Age	0.0349671	0.0143924	2.430	1.0400	0.02
Sex	0.7941428	0.3663038	2.168	2.2100	0.03
IHD	0.1554229	0.4968806	0.313	1.1700	n.s.
CKD	2.1506017	0.8105475	2.653	8.5900	0.008
Hypertension	0.5875029	0.3612523	1.626	1.8000	n.s.
CK (U/L)	0.0013924	0.0007198	1.934	1.0000	0.05

Deviance Residuals: Min, −2.1246; Median, −0.5129; Max, 2.2455. CKD, chronic kidney disease; IHD, ischemic heart disease; n.s., not significant.

## Data Availability

Anonymized study data can be requested by contacting Dr. Daniele Orsucci (orsuccid@gmail.com).

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
