# Peer review of "Increased Creatine Kinase May Predict A Worse COVID-19 Outcome"

_jcm, 2021, doi:10.3390/jcm10081734_

Round 1
Reviewer 1 Report
The manuscript by Orsucci et al followed a retrospective cohort design including 331 hospitalized COVID-19 patients classed as “mild” or “severe” from Italy. Clinical data collected at admission or follow-up was used to investigate whether serum creatine kinase levels were a predictor of disease severity when controlled for comorbidities and confounding factors. HyperCKemia was greater in severe than mild COVID-19 patients.
As the authors point out, CK has been identified as an independent and highly significant risk factor for mortality in COVID-19 patients in China e.g. (Zhang et al 2021; PLoSONE) and (Zhang 2020 J Thorac Dis). An aim of this study is “to confirm (or exclude) this relationship in a Caucasian population)”. However several such studies or case reports have already been published in Europe eg (Rivas-Garcia (2020) doi: 10.1093/rheumatology/keaa351; Pitscheider (2020) doi:10.1111/ene.14564) but this is not made clear in the introduction. It is not clear therefore, what if any new information this study brings.
No reference is given for the statement that “hyperCKemia: CK >200 U/L, which is a gen- erally accepted cut-off”. In fact, this seems to be quite low. Kyriakides (2010) report the 97.5th percentile in non-black females (217 IU/L) and non-black males (336 IU/L) is higher than this, and recommend a hyperCKemia cutoff 1.5x greater than these values.
No power calculations are provided so it is not possible to determine whether the study was suitably powered to detect differences in the 9 or so comorbidity outcomes.
Author Response
1) As suggested, we have specified in the Introduction that "Rare case reports from Europe are also available", and we cited the paper by Rivas-Garcia (2020). The article of Pitscheider (2020) is discussed and cited in the Discussion.
2) In the Methods, we have specified hyperCKemia: CK >200 U/L, "which is the upper limit of normal in our laboratory, not different from previous studies on COVID-19 [7]"
We are aware that the "reference values" of CK are a very debated issue, and for this reason we have also performed an analysis considering CK a continuous measure (irrespective of the arbitrary "normal" values") which confirmed the same findings.
3) As suggested, we have performed a sample size calculation (see Methods) which confirmed that our study was not under-powered.
Reviewer 2 Report
It is a retrospective cohort study of 331 consecutive COVID-19 patients who have 28 been hospitalized during the Italy’s “first wave” is to verify the relationship between CK levels and prognosis.
First of all, we do not know if it is a unicenter or multicenter. The sample size is smaller, so I would recommend that you clarify it well in the text, and if it is possible to expand the sample size to demonstrate causality.
In all the text, you mentione “severe” outcome of COVID-19, but It would be necessary to define what is a bad prognosis already in the abstract and describe the events considered extensively, because it is important.
Author Response
1) As suggested, we have specified (Abstract and Methods) that our study is "single-center".
2) We have performed a sample size calculation which confirmed that our study was not under-powered.
3) A requested, we have specified in the Abstract that "Severe disease is defined here as death and/or mechanical invasive ventilation, in contrast to “mild” patients who were discharged alive with no need for invasive ventilation; this latter group could also include those patients who were treated with non-invasive ventilation."
Reviewer 3 Report
Among other thinngs, it has been established that SARS-Cov-2 infection causes multi-organ failure. The authors found that it was also accompanied by an increase in muscular creatinine kinase (CK) especially in patients with severe course of this infection and some accompanying diseases. The weaknes of this study in the lack a more detailed clinical analysis of patients, including indicators of the efficiency of the respiratory system, heart and kidneys. The value of the work is icreased by such elements as the relatively large number of patients included in the study, and proper statistical analysis. Conclusion: The work has some clinical value, has been property prepared in terms and may be published in the presented version.
Author Response
We are very pleased of the comments of reviewer 3, which were substantially positive.
As suggested, we have further underscored in the Discussion that "a weakness of this work is the lack a more detailed clinical evaluation."
Round 2
Reviewer 1 Report
The authors have addressed my concerns
Reviewer 2 Report
the manuscript is now better